# Microstructural Analysis of Whole-Brain Changes Increases the Detection of Pediatric Focal Cortical Dysplasia

**DOI:** 10.3390/diagnostics15182311

**Published:** 2025-09-11

**Authors:** Xinyi Yang, Shuang Ding, Song Peng, Wei Tang, Yali Gao, Zhongxin Huang, Jinhua Cai

**Affiliations:** 1Department of Radiology, Children’s Hospital of Chongqing Medical University, Chongqing 400014, China; 2Department of Radiology, Chongqing Health Center for Women and Children, Chongqing 401147, China; 3Department of Radiology, Women and Children’s Hospital of Chongqing Medical University, Chongqing 401147, China

**Keywords:** focal cortical dysplasia, magnetic resonance imaging, diffusion tensor imaging, magnetization transfer imaging, voxel-based morphometry

## Abstract

**Purpose**: Focal cortical dysplasia (FCD) is a common developmental malformation disease of the cerebral cortex. Although mounting evidence has suggested that FCD lesions have variable locations and topographies throughout the cortex, few studies have explored consistencies in structural connectivity among different lesion types. In this study, we analyzed microscopic structural changes via lesion analysis and explored structural changes in nonlesion regions across the brain. **Methods**: Diffusion tensor imaging (DTI) and magnetization transfer imaging were used to compare FCD lesions and contralateral normal appearing gray/white matter (cNAG/WM). Voxel-based morphometry was calculated for 28 children with FCD and 34 sex- and age-matched healthy participants. DTI indices of the FCD and healthy control groups were analyzed via the tract-based spatial statistic method to evaluate the microstructure abnormalities of WM fiber tracts in individuals with FCD. **Results**: In terms of FCD lesions, compared with those of the cNAG, the fractional anisotropy (FA) values were decreased, and the mean diffusivity (MD), axial diffusivity (AD), and radial diffusivity (RD) values were increased; the magnetization transfer ratios were also decreased. In terms of whole-brain changes due to FCD, compared with the healthy control group, the FCD group showed a decrease in the volume of the right hippocampus and left anterior cingulate cortex. FCD patients had lower FA values, higher MD values, lower AD values, and mainly increased RD values in relation to WM microstructure. **Conclusions**: Microstructural abnormalities outside lesion regions may be related to injury to the epileptic network, and the identification of such abnormalities may complement diagnoses of FCD in pediatric patients.

## 1. Introduction

Focal cortical dysplasia (FCD) is the most common cause of difficult-to-treat epilepsy in children and is surgically treatable, manifesting as cortical gray matter (GM)–white matter (WM) blurring [1]. The diagnosis of FCD relies on pathological diagnosis [2], but in actual clinical practice, lesions detected via routine magnetic resonance imaging (MRI) are important preoperative assessment criteria. Some cortical developmental malformations can be difficult to identify and can remain undetectable, even in high-field MR images. Subtle lesions may be missed in up to 30% of surgical candidates, resulting in them being incorrectly labeled as “MRI negative” or “nonlesion”, thereby missing the opportunity for surgery and exacerbating their drug resistance [3]. The accurate identification and assessment of lesions prior to surgery is key to increasing surgical opportunities and improving surgical outcomes [4,5].

MRI is the most commonly used noninvasive neuroimaging method for assessing the brain. High-field strength, high-resolution imaging sequences, and specific image preprocessing techniques can be used to further increase the sensitivity and specificity when identifying small lesions. To our knowledge, blurred GM–WM boundaries are the most prominent feature of FCD in MR images. Therefore, previous studies have used this approach as a starting point to study GM, WM, and lesions divided by GM–WM boundaries [6,7]. Some FCD patients have small lesions but severe cognitive impairment and behavioral abnormalities [8], suggesting that, in addition to studies based on cortical volume, clearer visualization of the lesion surroundings and exploration of the whole-brain structural network associated with the lesion are highly important.

Diffusion tensor imaging (DTI) can be used to reveal the microscopic structural integrity of tissue, particularly the orientation and integrity of WM fiber tracts. In DTI, the anisotropic diffusion of water molecules within the tissue is measured. Magnetization transfer imaging (MTI) indirectly reflects the microscopic structural integrity of tissue through detection of the magnetic resonance transfer effects of macromolecules (such as proteins and lipids) on free water in the tissue, particularly changes in myelin and cell membranes. DTI is more sensitive to axonal damage, whereas the magnetization transfer ratio (MTR) is more specific to changes in myelin. In addition, in WM with a normal appearance, a reduced MTR may be associated with axonal degeneration [9]. Multiangle detection of lesion microstructures using DTI and the MTR has been used in the case of some central demyelinating diseases [10,11]. The two techniques complement each other and can provide more comprehensive information on tissue microstructure and biochemistry. Previous studies have shown that microscopic structural abnormalities in children with FCD are not only present around the lesions but may also exist in WM distant from the lesions [12,13,14]. Tract-based spatial statistic (TBSS) analysis based on DTI is an automated whole-brain voxel-by-voxel tool that better reflects potential WM-related imaging markers unrelated to lesions throughout the brain [15]. Previous studies have investigated the sensitivity of lesions with blurred GM–WM boundaries. Furthermore, layer-by-layer studies of whole-brain GM volume and whole-brain WM networks distant from lesions have been performed. The results of these investigations have significant implications for understanding central nervous system disorders. However, their implications in cases of typical children with FCD remain unclear.

We hypothesize that focal abnormalities affect the whole brain, that heterogeneous lesions can cause similar symptoms, and that the connectivity pattern in individuals with FCD may be associated with whole-brain network topology. In this study, we investigated four DTI indices and MTR indices of lesions and assessed whole-brain changes in cortical volume and WM fiber tracts in pediatric FCD patients. The aim of this study was to explore the value of these indicators as imaging markers for assessing the microscopic structure of FCD lesions and to confirm similar damage at the whole-brain network level to clarify the widespread distribution of epileptic networks.

## 2. Materials and Methods

### 2.1. Participants

This study was approved by the Institutional Review Board of the Children’s Hospital of Chongqing Medical University (2022 Grant No. 161), and written informed consent was obtained from each participant prior to MRI scanning.

The study included a total of 28 children with FCD. All participants were recruited between July 2021 and December 2023. The inclusion criteria for children with FCD were as follows: (a) diagnosed with FCD according to standard guidelines [16] or postoperative pathology; (b) younger than 18 years of age; (c) no history of other neurological disorders. The exclusion criteria were as follows: (a) incomplete clinical data; (b) poor image quality; (c) recurrence of lesions or prior treatment history; (d) contraindications for MRI scanning. During the same period, another group of healthy control subjects matched with the FCD group in terms of age and sex was recruited, comprising 34 individuals. The same exclusion criteria as used with the FCD group was applied to the healthy control group.

### 2.2. MRI Protocol

All pediatric patients were scanned using a 3.0 T MRI scanner (Philips Achieva, Amsterdam, The Netherlands) with an 8-channel head coil. Bandages and foam provided by Philips were used to secure the head and reduce head movement during the scanning process. Earplugs were used to reduce interference from the noise emitted by the MRI machine during the scanning process.

Standard cranial MRI sequences: axial T2-weighted imaging (T2WI) using fast gradient echo sequence: TR = 3500 ms, TE = 80 ms; Axial T2-FLAIR sequence: TR = 8000 ms, TE = 125 ms; axial T1-weighted imaging (T1WI) using fast inversion recovery (IR) sequence, TR = 2000 ms, TE = 20 ms. All axial fields of view (FOV) are 230 mm × 191 mm × 143 mm, slice thickness = 5 mm, an interval of 1 mm, imaging time 7 min 20 s. 3D-T1WI structural sequence: Acquired using a phase-encoded gradient echo sequence with TR = 7.4 ms, TE = 3.8 ms, slice thickness 1 mm, slice spacing 0 mm, 260 slices, FOV = 250 mm × 250 mm × 156 mm, scan matrix 228 × 227, number of repetitions 1, flip angle 8°, voxel size = 0.60 mm × 1.04 mm × 1.04 mm, imaging time 4 min 16 s. DTI sequence acquired using a spin-echo-planar echo sequence: TR = 7865 ms, TE = 66 ms, flip angle 90°, voxel size = 2 mm × 2 mm × 2 mm, slice thickness 2 mm, slice spacing 0 mm, 70 slices, FOV 224 mm × 224 mm × 140 mm, scan matrix 112 × 112, diffusion sensitivity coefficient b = 1000 s/mm^2^, imaging time 5 min 41 s. MTI sequence: TR = 84 ms, TE = 2.7 ms, slice thickness 2 mm, slice spacing 0 mm, 160 slices, FOV 224 mm × 224 mm × 160 mm, scan matrix 224 × 224, flip angle 18°, voxel size = 1.5 mm × 1.5 mm × 2 mm, imaging time 9 min 04 s.

### 2.3. Calculation of MRI Indicators

#### 2.3.1. DTI and MTI Indicators Based on Lesions

The DTI processing steps were as follows: (a) image format conversion; (b) eddy current correction; (c) skull stripping to remove nonbrain tissue; (d) tensor calculation to obtain fractional anisotropy (FA), mean diffusivity (MD), axial diffusivity (AD), and radial diffusivity (RD) maps.

The MTR image is a percentage image obtained from an MR image obtained without using pulses and an MR image obtained using saturated pulses. These images were processed in FSL (version: 5.0.9, www.fmrib.ox.ac.uk/fsl (accessed on 1 September 2021)) software using the following formula:$$MTR=\frac{\left(M0-M1\right)}{M0}*100$$
where M0 is the signal intensity of pixels obtained without magnetization transfer pulses, and M1 is the signal intensity of pixels obtained with magnetization transfer pulses. The processed MTR images were obtained after skull stripping.

The DTI parameters and processed MTR image were simultaneously registered to a 3D-T1WI image and then loaded into ITK-SNAP software (version 4.0.0, http://www.itksnap.org (accessed on 12 September 2021)) in NII format. Two experienced radiologists manually outlined the three-dimensional volume of the lesion in a layer-by-layer manner along the lesion contour on the 3D-T1WI images via a double-blinded process. The images of the participants in the normal control group were automatically flipped to the cNAG/WM regions according to a mirror symmetry approach. Intraclass correlation coefficients (ICCs) were used to assess intraobserver and interobserver consistency, with an ICC > 0.75 considered to indicate good consistency. The lesion volume, FA values, MD values, AD values, RD values, and MTR values were then extracted from the lesions.

#### 2.3.2. Voxel-Based Morphometry (VBM) Indicators

The 3D-T1WI data were processed using the VBM8 toolbox in SPM8 (University College London, UK) in MATLAB 2013b (MathWorks, Natick, MA, USA). The process included the following steps: (a) remove images with scan artifacts and anatomical abnormalities; (b) align image planes to the anterior commissure (AC) and posterior commissure (PC) lines on the sagittal plane; (c) segment the images into GM, WM, and cerebrospinal fluid and perform registration/normalization using DARTEL registration technology; (d) convert the images to Montreal Neurological Institute (MNI) space; (e) smooth the GM images using a Gaussian kernel with a full-width at half-maximum (FWHM) of 8 mm.

#### 2.3.3. TBSS Based on DTI

DTI data were preprocessed using the FMRIB Software Library (FSL, version 5.0.9; http://www.fmrib.ox.ac.uk/fsl (accessed on 15 October 2021)). The preprocessing steps were as follows: (a) format conversion; (b) head motion correction; (c) obtain the brain mass; (d) calculate the DTI parameters (FA, AD, RD, MD) on the basis of the DTI data. TBSS analysis was subsequently performed as follows: (a) convert the FA images; (b) perform nonlinear registration; (c) extract the skeleton; (d) perform a skeleton projection to obtain the FA skeleton images; (e) design the contrast; (f) perform statistical analysis using permutation testing; (g) display the expanded visualization results. The same processing method was used for the MD, AD, and RD parameters between the two groups of participants.

### 2.4. Statistical Analysis

All statistical analyses were performed using SPSS (version 26.0). Continuous variables were analyzed using independent-sample *t*-tests, with data expressed as the means ± standard deviations (SDs) or medians (interquartile ranges, IQRs). Categorical variables were analyzed using chi-squared tests, with data expressed as absolute numbers and percentages. Independent-sample *t*-tests were used to analyze lesion volume; FA, MD, AD, and RD values; and MTR values, with age and sex as covariates. In the VBM analysis, general linear models (GLMs) in SPM8 were used to analyze the data and assess differences (*p* < 0.05, FDR correction, cluster > 20 voxels). In the TBSS analysis, threshold-free cluster enhancement (TFCE) and familywise error (FWE) corrections were applied for multiple comparisons. *p* < 0.05 was considered to indicate a significant difference.

## 3. Results

### 3.1. General Clinical Data and Lesion Distribution

The general clinical data and lesion distributions of the 28 patients in the FCD group and the 34 individuals in the healthy control group are shown in Table 1. There were no statistically significant differences between the FCD group and the healthy control group in terms of sex, age, or right-handedness (*p* > 0.05). The lesions were most commonly distributed in the frontal lobe (39%), followed by the parietal lobe (32%).

### 3.2. DTI and MTI Results Based on Lesions of Pediatric FCD

Figure 1 shows the T1WI, MTI, and four DTI parameter maps of a child with FCD. The lesion was located in the right parietal lobe, with imaging results showing local cortical thickening and blurred GM–WM boundaries. The lesion volume and cNAG/WM volume were not significantly different. The four DTI parameter values were significantly different between the lesion volume and cNAG/WM volume. The FA and MTR values decreased in the lesion area compared with those in the control area, whereas the MD, RD, and AD values increased. The FA values in the lesion region and the contralateral control area were 0.226 ± 0.056 and 0.259 ± 0.052, respectively (*p* < 0.05), while the MD values were 0.922 ± 0.092 and 0.868 ± 0.074 (*p* < 0.05), the RD values were 0.823 ± 0.088 and 0.765 ± 0.094 (*p* < 0.05), and the AD values were 1.160 ± 0.089 and 1.080 ± 0.043 (*p* < 0.05). The MTR values in the lesion and contralateral control areas were 0.570 ± 0.205 and 0.817 ± 0.201, respectively (*p* < 0.05). The detailed results are shown in Table 2 and Figure 2.

### 3.3. VBM Results Based on the Whole Brain of Pediatric FCD Patients

Compared with those in the normal control group, the volumes of the right hippocampus and left anterior cingulate cortex were lower in the FCD group (Table 3 and Figure 3).

### 3.4. TBSS Results Based on the Whole Brain of Pediatric FCD Patients

The results of the TBSS analysis indicate that individuals with FCD displayed widespread decreases in FA and AD values across whole-brain white-matter fibers, as well as increases in MD values and RD values. However, there were also contradictory decreases in RD values in some regions.

The regions with reduced FA values in the FCD group compared with the healthy control group included the middle cerebellar peduncle, left cingulate gyrus, and left superior corona radiate (Table 4 and Figure 4). The regions with increased MD values in the FCD group compared with the healthy control group included the middle cerebellar peduncle, right internal capsule, bilateral posterior thalamic radiation, bilateral medial lemniscus, bilateral external capsule, bilateral superior longitudinal fasciculus, left cerebral peduncle, bilateral superior corona radiate, left fornix, and left cingulate gyrus (Table 5 and Figure 4). The regions with increased RD values in the FCD group compared with the healthy control group included the bilateral hippocampus, bilateral superior corona radiate, left anterior corona radiate, right cingulate gyrus, and splenium of the corpus callosum. The regions with decreased RD values in the FCD group compared with the healthy control group included the middle cerebellar peduncle, left anterior corona radiate, left fornix, left genu of the corpus callosum, right cerebral peduncle, sagittal stratum, and bilateral external capsule (Table 6 and Figure 5). The regions with reduced AD values in the FCD group compared with the healthy control group included the left anterior corona radiate, bilateral hippocampus, right superior corona radiate, left cingulate gyrus, and splenium of the corpus callosum (Table 7 and Figure 4).

## 4. Discussion

In this study, lesion-based DTI, MTR indices, and voxel-based whole-brain connectivity analysis were integrated to confirm the widespread impact of focal FCD lesions on the whole-brain network, especially WM fiber tracts. These findings provide imaging evidence showing that focal abnormalities cause systemic dysfunction throughout the brain, which may explain why FCD patients with high pathological heterogeneity (such as types IIa and IIb) exhibit similar seizure phenotypes (such as drug-resistant focal seizures).

### 4.1. Lesion-Related Microstructural Damage in Individuals with FCD

Cortical development is closely related to WM, so developmental abnormalities in individuals with FCD may also involve deeper WM tissues. Microstructural analysis based on myelin damage has been performed to detect subtle, visually elusive lesions in individuals with FCD. This study is the first in which DTI and the MTR were combined to study FCD in a pediatric population, confirming their synergistic role in assessing microstructural damage associated with lesions. The results show that the FA, MD, RD, AD, and MTR values were significantly different between the lesion and contralateral control regions. The FA and MTR values decreased in the lesion area compared with those in the control area, whereas the MD, RD, and AD values increased. These results suggest that DTI indices and MTR values may be potential biomarkers for assessing microscopic damage in individuals with FCD. The reduction in the MTR values further confirms the microscopic structural abnormalities in FCD lesion areas, which may reflect a reduction in myelin or macromolecular protein content. The difference in DTI indices between the FCD lesion areas and cNAG/WM regions is attributed mainly to the higher diffusion rate of water molecules in FCD lesion areas, which is due to changes in the microscopic structure of WM, including myelin loss, abnormal myelin formation, neuronal death, dendrite reduction, and reactive gliosis caused by recurrent episodes [12]. FA is the most commonly used DTI metric and serves as a marker of axonal integrity, exhibiting high sensitivity to the microscopic structural integrity of fibers [17]. MD is sensitive to cellular edema and necrosis [10], AD is associated with axonal damage [18], and RD is a myelin marker [19], with all three indicators providing specific complementary information to FA values, reflecting the diffusion state of tissue molecules from different perspectives. Compared with traditional MRI, MTI is a more sensitive approach for the early detection of Wallerian degeneration along anatomical pathways [20], and the number of protons involved in pathological processes associated with demyelination and axonal damage is decreased in this case. In this study, the MTR values at lesion sites were significantly lower than those at cNAG/WM sites, and we speculate that this may be due to neuronal loss, axonal injury, demyelination, and disruption of microstructural integrity, which is consistent with the brain tissue changes reflected by the DTI indices. Furthermore, these two indicators corroborate each other, significantly enhancing the detection rate of FCD microlesions.

### 4.2. Whole-Brain Cortical Volume Damage in Individuals with FCD

The results indicate that the volumes of the right hippocampus and left precuneus in children with FCD were significantly lower than those in healthy controls, yet these areas did not correspond to regions with apparent lesions. This is not a contradiction but rather strong evidence that FCD is a whole-brain network disorder. FCD lesions exert a “remote effect” through neural connections, influencing the structure and function of distant brain regions (the critical hubs for cognition such as the hippocampus and anterior cingulate cortex), leading to the clinical cognitive impairments observed in patients.

The most important function of the hippocampus is encoding spatial memory, and damage to this region can lead to memory impairment [21]. Previous studies have hypothesized that epilepsy-related toxic secretions are not limited to the epileptic focus but rather spread to other brain regions, and that the local excitotoxic effects of the spread of epileptic activity may lead to neuronal loss [22,23,24,25]. In this study, the right hippocampal volume was reduced in the FCD group compared with that in the healthy control group, which is speculated to be related to the susceptibility of the hippocampus, a critical brain region, to epileptic damage. Although the lesion locations varied in this study, the epileptogenic substances transmitted through the epileptic network from epileptogenic foci all caused hippocampal damage. Frequent damage leads to neuronal degeneration and destruction, ultimately resulting in a reduction in hippocampal volume. In addition, anatomical and connectivity studies have shown that the anterior cingulate cortex is an important node in a broad network involving cortical and subcortical regions [26,27]. Structural abnormalities in the precuneus may affect higher-order cognitive functions, such as self-referential processing and episodic memory.

However, there were no significant differences in lesion volume between the FCD regions and the cNAG/WM regions, suggesting that localized lesions in individuals with FCD may not lead to widespread atrophy among GM regions. The functional implications of these findings require further investigation.

### 4.3. Whole-Brain WM Microstructural Damage in Individuals with FCD

Additionally, compared with healthy controls, FCD patients presented lower FA values, higher MD values, and more pronounced increases in RD values across various brain regions, indicating that WM damage is not limited to the lesion area but may involve broader neural networks. These findings are consistent with the notion that FCD involves the disruption of cortico-WM connections, indicating that its pathophysiology might not be restricted to the cortex alone. Significant decreases in FA values and increases in MD values within the lesion area are consistent with possible axonal disruption and edema in these regions. Additionally, observed changes in FA and MD values in distant regions (such as the corpus callosum-pressurized region of the contralateral hemisphere) may suggest that white matter damage could spread along connecting fibers. Gennari et al. [28] found that DTI-based extension techniques can be used to observe differences in FA and MD values between FCD lesions and the contralateral brain parenchyma. Our findings are consistent with the idea of epileptic network hemispheric reorganization proposed in previous studies.

Focal structural abnormalities in individuals with FCD may affect the entire brain through two pathways. The first potential pathway involves diffuse structural changes. It is hypothesized that axonal damage in the lesion area could result in Wallerian degeneration of downstream fibers. This interpretation is consistent with the finding of reduced fractional anisotropy (FA) in the corticospinal tract on DT images. Abnormalities in “highway” structures like the internal capsule, corona radiata, and corpus callosum facilitate easier seizure generalization, making drug control more challenging. Damage to the longitudinal fasciculus, cingulate gyrus, and hippocampal microstructure directly disrupts the neural networks supporting higher cognitive functions such as learning and memory. Abnormal pathways in the internal capsule, cerebral peduncles, and cerebellar peduncles may explain difficulties with fine motor skills. Extensive involvement of the limbic system (cingulate gyrus, hippocampus, and fornix) is closely associated with emotional regulation disorders. Notably, increased MD was the most widespread abnormality in this study, indicating that demyelination and axonal injury have already caused macroscopic structural loosening. Moreover, reduced AD and increased RD observed in the hippocampus and corpus callosum suggest a pattern of axonal injury, potentially due to Wallerian degeneration or developmental loss caused by abnormal neural network activity. In addition, the superior corona radiata and cingulate gyrus showed insignificant AD changes but markedly increased RD, suggesting a pattern of demyelinating injury. Changes in the microscopic structure of fiber bundles may be driven by the spread of seizures rather than being the cause of seizures [29]. Urquia-Osorio et al. [30] evaluated surface and deep WM diffusion abnormalities in patients with focal epilepsy (FCD is an important component of the etiology of this disease) via DTI. Previous studies have reported microscopic structural changes in various WM regions, including the corpus callosum, cingulate gyrus, and internal and external capsules, in children with epilepsy [31,32,33,34,35].

The second potential pathway involves functional compensatory reorganization. Unaffected regions could potentially generate hypersynchronous discharge, possibly mediated by an increase in synaptic connections. This RD discrepancy may also result from abnormal neurodevelopment causing an abnormally dense arrangement of fiber bundles. These two pathways may provide a potential explanation for the contradictory results in respect of both increased and decreased RD values.

This study has several limitations. First, the small sample size and wide age range may limit the statistical power of the neurocognitive assessment results. These findings could be validated in future studies by increasing the sample size to obtain age-appropriate neurocognitive assessments and exploring their relationship with imaging biomarkers, or by combining various functional imaging techniques to provide clearer insights for clinical management. Second, analysis of the associations between WM abnormalities and seizures or cognitive impairments requires long-term follow-up studies. In reality, the sensitivity of DTI to axonal damage may be affected by partial volume effects (such as blurred lesion boundaries), whereas the interference of glial proliferation must be excluded owing to the specificity of the MTR to macromolecular changes. Third, histological verification should be included in future studies.

## 5. Conclusions

In this study, DTI and the MTR were used to assess microscopic structural changes in FCD lesions. Then, whole-brain GM and WM structural changes were explored via VBM and TBSS analysis in children with FCD, revealing that FCD not only affects cortical development but may also be accompanied by widespread WM microstructural changes. This study provides new insights into the neuropathological mechanisms underlying FCD, indicating that changes in WM microstructure may be an important biological feature of this disease and that such changes could serve as targets for diagnosis and treatment in the future.

## Figures and Tables

**Figure 1 diagnostics-15-02311-f001:**
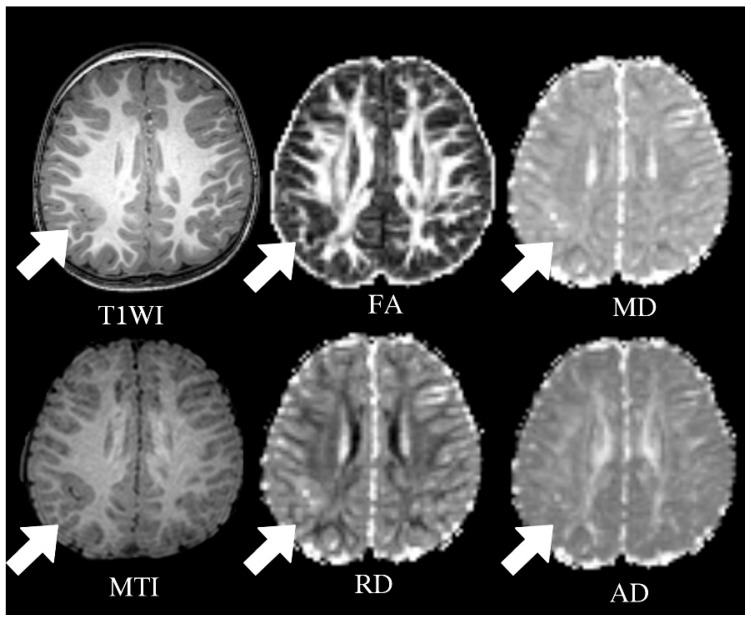
Multimodal MRI parametric map of a child with FCD whose lesion was located in the right parietal lobe, showing local cortical thickening and unclear boundary between gray and white matter. The arrow indicates the lesion of FCD. FA: fractional anisotropy; MD: mean diffusivity; AD: axial diffusivity; RD: radial diffusivity; MTI: magnetization transfer imaging.

**Figure 2 diagnostics-15-02311-f002:**
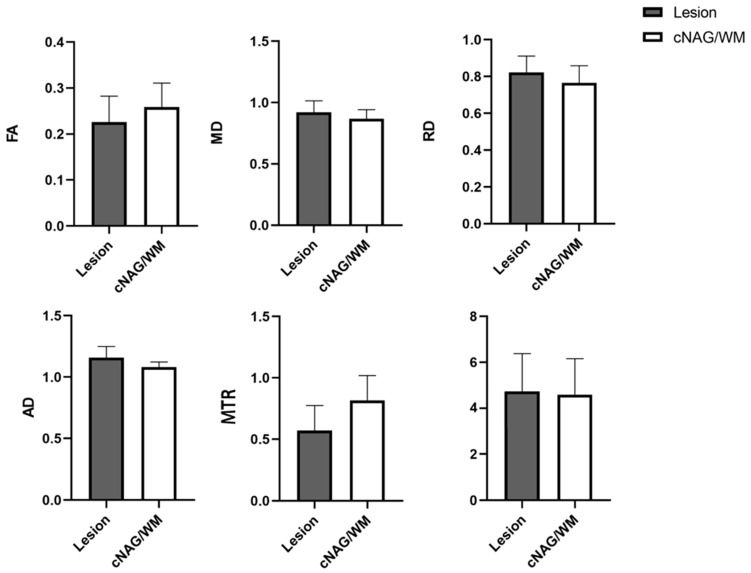
Statistical data in respect of FA, MD, RD, AD, MTR, and volume for different FCD and cNAG/WM. FCD: focal cortical dysplasia, cNAG/WM: contralateral normal appearing gray/white matter.

**Figure 3 diagnostics-15-02311-f003:**
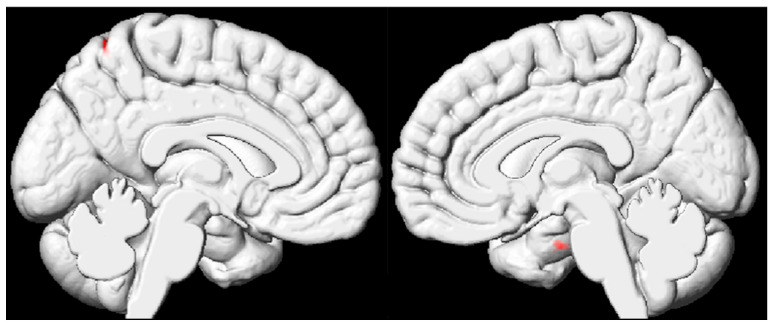
Brain regions with different volume changes based on whole brain voxel level. Brain regions with reduced volume are marked in red, designated as the right hippocampus and left anterior cingulate cortex.

**Figure 4 diagnostics-15-02311-f004:**
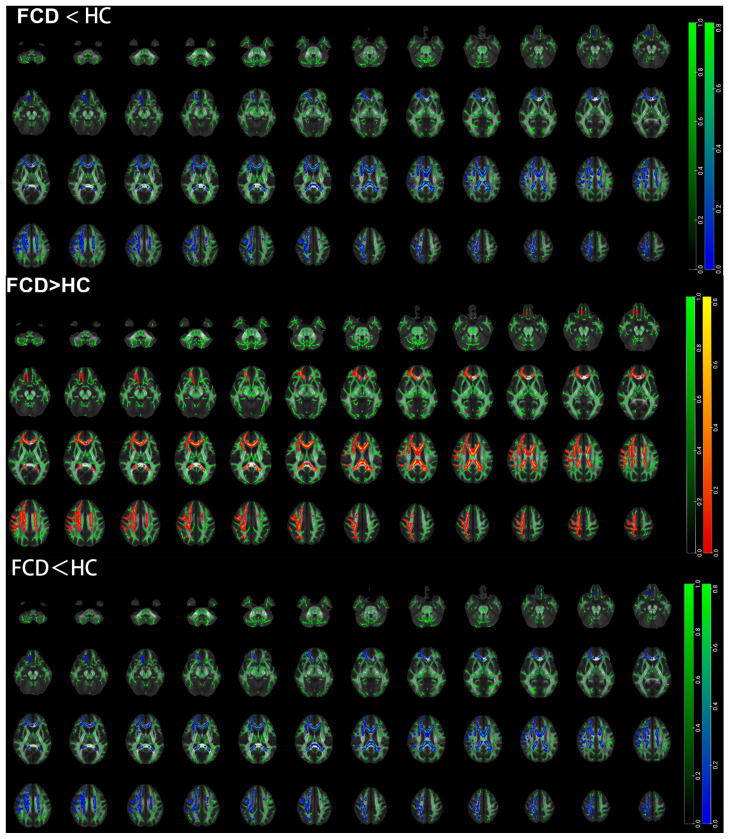
Differences in FA, MD, and AD values between FCD and HC groups (*p* < 0.05, TFCE corrected). Green: white-matter skeleton; blue: regions with reduced value; red–yellow: regions with increased value; FCD: focal cortical dysplasia; HC: healthy control.

**Figure 5 diagnostics-15-02311-f005:**
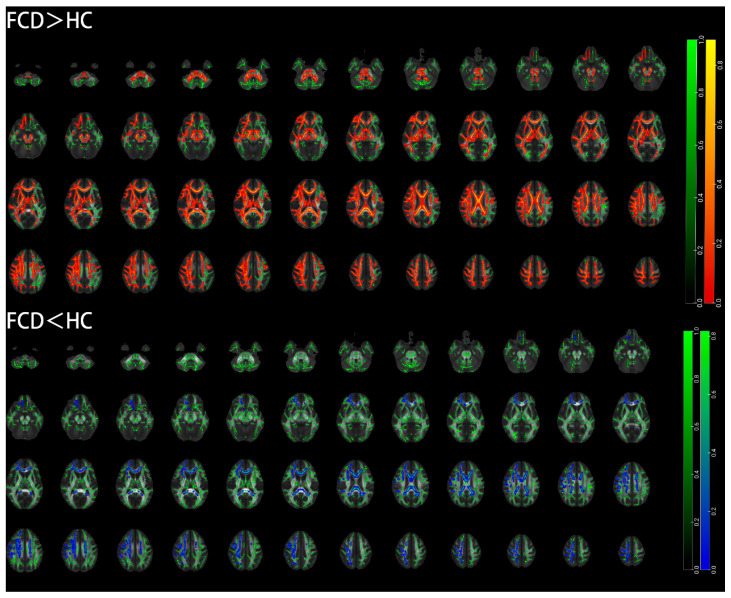
Differences in RD values between FCD and HC groups (*p* < 0.05, TFCE corrected). Green: white-matter skeleton; blue: regions with reduced value; red–yellow: regions with increased value; FCD: focal cortical dysplasia; HC: healthy control.

**Table 1 diagnostics-15-02311-t001:** Clinical information in respect of participants included in the study and lesion location of FCD.

	FCD (*n* = 28)	Healthy Controls (*n* = 34)	*p*
Sex (male/female)	14/14	21/13	0.999
Age	8 (5, 11.5)	6 (4.75, 7)	0.999
Age of seizure onset	7 (5, 9.75)	NA	
Family history	15 (54%)	NA	
Right-handed	23 (82%)	30 (88%)	0.999
Distribution of lesions (by brain lobe)		NA	
Frontal lobe	11 (39%)		
Parietal lobe	9 (33%)		
Occipital lobe	3 (11%)		
Temporal lobe	3 (11%)		
Insula	2 (7%)		

Note: Data are presented as median and interquartile range or n (%). FCD: focal cortical dysplasia.

**Table 2 diagnostics-15-02311-t002:** The difference in imaging indicators between FCD and cNAG/WM.

	FCD	cNAG/WM	t	*p*
Lesion volume (mm^3^)	4.737 ± 1.638	4.588 ± 1.567	0.316	0.753
FA	0.226 ± 0.056	0.259 ± 0.052	−2.114	0.040 *
MD	0.922 ± 0.092	0.868 ± 0.074	2.191	0.034 *
RD	0.823 ± 0.088	0.765 ± 0.094	2.160	0.036 *
AD	1.160 ± 0.089	1.080 ± 0.043	3.906	0.000 *
MTR	0.570 ± 0.205	0.817 ± 0.201	−4.130	0.000 *

Note: Data are represented as mean ± standard deviation. FCD: focal cortical dysplasia, cNAG/WM: contralateral norma appearing gray/white matter, FA: fractional anisotropy, MD: mean diffusivity, AD: axial diffusivity, RD: radial diffusivity, MTR: magnetization transfer ratio, *: significant difference.

**Table 3 diagnostics-15-02311-t003:** Brain regions with different volume changes based on whole brain voxel level.

Brain Region (AAL)	Cluster Size (Voxel)	Peak MNI Coordinates			t Value
		x	y	z	
FCD < HC					
Cluster 1					
Right hippocampus	26	19.5	−9	33	−4.4148
Cluster 2					
Left precentral gyrus	20	−13.5	−61.5	−64.5	−5.1828

Note: MNI, Montreal Neurological Institute; AAL, automated anatomical labeling; FCD: focal cortical dysplasia. *p* < 0.05, FDR corrected, cluster > 20 voxels.

**Table 4 diagnostics-15-02311-t004:** Differences in FA values between FCD patients and healthy controls.

Brain Region (AAL)		Cluster Size (Voxel)	Peak MNI Coordinates			t Value
			x	y	z	
FCD < HC						
Middle cerebellar peduncle		115	24.3	−60.4	−38.8	0.965
Cingulum (cingulate gyrus)	L	50	−7.46	19.7	25.1	0.971
Superior corona radiata	L	10	−28.4	−0.197	19.2	0.971

Note: MNI, Montreal Neurological Institute; AAL, automated anatomical labeling; FCD: focal cortical dysplasia. *p* < 0.05, TFCE corrected, cluster > 10 voxels.

**Table 5 diagnostics-15-02311-t005:** Differences in MD values between FCD patients and healthy controls.

Brain Region (AAL)		Cluster Size (Voxel)	Peak MNI Coordinates			t Value
			x	y	z	
FCD > HC						
Middle cerebellar peduncle		95	−12.2	−22.1	−28.4	2.1
Internal capsule	R	75	36.2	−37	8.62	1.98
Posterior thalamic radiation	R	57	35.2	−55.6	0.364	1.82
Medial lemniscus	L	42	−4.15	−34.8	−40.3	2.43
External capsule	L	34	35.5	−56.9	13.7	1.87
Superior longitudinal fasciculus	R	26	40.7	−40.8	17	1.54
Cerebral peduncle	L	25	−11.2	−26.5	−17.8	2.12
Posterior thalamic radiation	L	25	−31.2	−67	7.41	2.22
Medial lemniscus	R	16	4.51	−37.1	−34.9	1.92
Superior corona radiata	R	15	27.4	−23.2	33.2	2.06
Fornix	L	14	−25.7	−33.2	7.25	1.98
External capsule	R	11	35.1	−2.74	−11.6	1.67
Superior corona radiata	L	10	−21	−14.6	35.5	1.37
Superior longitudinal fasciculus	L	10	−35.9	0.816	26	1.4
Cingulum (cingulate gyrus)	L	10	−6.44	−9.68	35.9	1.8

Note: MNI, Montreal Neurological Institute; AAL, automated anatomical labeling; FCD: focal cortical dysplasia. *p* < 0.05, TFCE corrected, FWE: family-wise error, cluster > 10 voxels.

**Table 6 diagnostics-15-02311-t006:** Differences in RD values between FCD patients and healthy controls.

Brain Region (AAL)		Cluster Size (Voxel)	Peak MNI Coordinates			t Value
			x	y	z	
FCD > HC						
Cingulum (hippocampus)	L	27	−22.4	−29.6	−16.7	1.76
Superior corona radiata	L	20	−31.4	−10.2	23.1	1.64
Superior corona radiata	R	14	27.4	−23.2	33.2	3.14
Anterior corona radiata	L	12	−20.9	28.1	21.3	1.77
Cingulum (cingulate gyrus)	L	12	−8	32.6	14.1	2.64
Cingulum (hippocampus)	R	11	26.1	−20.3	−26.5	1.29
Splenium of corpus callosum		11	15.5	−45.2	28.6	2.36
FCD < HC						
Middle cerebellar peduncle		38	−22.6	−62	−11.2	2.2
Anterior corona radiata	L	18	−17.6	28	−11.2	1.4
Fornix	L	23	−30.8	−28.1	−2.82	1.48
Genu of corpus callosum		13	−12	21.2	−6.51	1.93
Cerebral peduncle	R	12	12	−23.8	−10	1.6
Sagittal stratum	L	11	−35.5	−19.5	−6	1.02
External capsule	R	10	35	4	−3	1.07
External capsule	L	10	−34	−1	−2	1.07

Note: MNI, Montreal Neurological Institute; AAL, automated anatomical labeling; FCD: focal cortical dysplasia. *p* < 0.05, TFCE corrected, cluster > 10 voxels.

**Table 7 diagnostics-15-02311-t007:** Differences in AD values between FCD patients and healthy controls.

Brain Region (AAL)		Cluster Size (Voxel)	Peak MNI Coordinates			t Value
			x	y	z	
FCD < HC						
Anterior corona radiata	L	32	−20.9	28.1	21.3	1.77
Cingulum (hippocampus	L	27	−22.4	−29.6	−16.7	1,76
Superior corona radiata	R	14	27.4	−23.2	33.2	3.14
Cingulum (cingulate gyrus)	L	12	−8	32.6	14.1	2.64
Cingulum (hippocampus)	R	11	26.1	−20.3	−26.5	1.29
Splenium of corpus callosum		11	15.5	−45.2	28.6	2.36

Note: MNI: Montreal Neurological Institute; AAL: automated anatomical labeling; FCD: focal cortical dysplasia. *p* < 0.05, TFCE corrected, cluster > 10 voxels.

## Data Availability

Data cannot be shared openly to protect study participant privacy.

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
