# Peer review of "Microstructural Analysis of Whole-Brain Changes Increases the Detection of Pediatric Focal Cortical Dysplasia"

_diagnostics, 2025, doi:10.3390/diagnostics15182311_

Round 1

Reviewer 1 Report

Comments and Suggestions for Authors

Dear Professor

I have carefully reviewed the manuscript submitted to Diagnostics. This study is grounded in a valuable conceptual framework—namely, the perspective that focal cortical dysplasia (FCD) should be regarded as a disease involving not only localized lesions but also widespread alterations in whole-brain microstructural networks. From the viewpoint of a clinical epileptologist, this direction is both timely and meaningful.

In particular, the authors’ effort to capture widespread white matter changes beyond the lesion area through combined DTI and MTI analysis is commendable and novel. This attempt aligns with the journal’s focus and purpose. It also has the potential to supplement conventional MRI in detecting subtle or MRI-negative FCD cases, or in patients whose seizure semiology does not precisely match the identified lesion. As such, this work may contribute to future advances in presurgical evaluation and diagnosis.

With regard to ROI placement, the use of manual segmentation by two experienced radiologists in a blinded fashion is, at present, an acceptable standard given the limitations of current imaging techniques. The authors' methodology is generally reasonable in this context.

However, from a clinical translational perspective, several limitations remain.

First, while the authors discuss the clinical utility of this technique, the imaging technology may not yet be sufficient to directly influence surgical decision-making or pathologic interpretation in epilepsy surgery. It is important to recognize that the sample size is limited and no stratification was performed by seizure semiology, lesion location, or FCD pathological subtype (e.g., type I, IIa, IIb). Therefore, the generalizability of the findings to the broader FCD population is uncertain, and the study cannot yet support direct clinical implementation. Additionally, the manuscript does not address neurocognitive findings in FCD patients, which would have strengthened its relevance to the clinical management of epilepsy.

Second, the integration of figures and text could be improved. Figures 2–5 and Tables 2–7 are not sufficiently explained in the main text, and non-specialist readers may find them difficult to interpret. Clarifying their clinical and scientific relevance would greatly enhance readability.

Third, the authors discuss Wallerian degeneration and global network reorganization in a rather assertive tone, despite the study being cross-sectional in design. These are hypothetical phenomena and should be described more cautiously, using expressions such as "may suggest" or "is potentially consistent with." Such changes are not directly verifiable in a single time-point analysis and should be framed as working hypotheses, not conclusions.

Lastly, I note several typographical errors that should be corrected before publication:

Line 238–239: “Superiorr”

Line 252–253: “bule”

In summary, this manuscript proposes an important and forward-looking concept, and with appropriate revisions and cautious reframing of some interpretations, it has the potential to make a valuable contribution to the field of pediatric epilepsy imaging.

Best regards,

Reviewer

Reviewer 2 Report

Comments and Suggestions for Authors

This is a well written and sophisticated study. It suggests that we have previously underestimated the extent of brain change in children with focal cortical dysplasia. The findings are very persuasive. It appears that the FCD lesions are just the tip of the iceberg of brain abnormalities. The imaging data supports the authors' contention that FCD lesions are associated with Wallerian degeneration and widespread network dysfunction. This is a cautionary tale: circumscribed excision of lesions may not significantly overcome the developmental problems.

Round 2

Reviewer 1 Report

Comments and Suggestions for Authors

Dear Authors

Regarding our Comment 1, the authors generally responded as follows: Concerning neurocognitive function, due to the wide age range of the participants and the lack of a unified neurocognitive assessment battery applicable to all pediatric age groups, no such evaluation was performed in this study. In the initial stage, cognitive assessments were conducted in a subset of patients, but no significant correlations with imaging findings were observed. Therefore, these results were not included in the manuscript. In future studies, age-appropriate neurocognitive assessments will be incorporated, and their relationship with imaging biomarkers will be examined to provide clearer insights for clinical management. We recommend adding this point to the Discussion section as a limitation of the current study and as a reference for future research.

Regarding the other some points, the authors have sincerely addressed all of my questions. This manuscript is novel, and we look forward to the development of future studies.

Thank you for your submission for Diagnositics.

Best regards, 

reviewer
